# Caffeine: A Multifunctional Efficacious Molecule with Diverse Health Implications and Emerging Delivery Systems

**DOI:** 10.3390/ijms252212003

**Published:** 2024-11-08

**Authors:** Xinjie Song, Mahendra Singh, Kyung Eun Lee, Ramachandran Vinayagam, Sang Gu Kang

**Affiliations:** 1Zhejiang Provincial Key Lab for Chemical and Biological Processing Technology of Farm Product, School of Biological and Chemical Engineering, Zhejiang University of Science and Technology, Hangzhou 310023, China; foodtech.song@foxmail.com; 2Department of Biotechnology, Institute of Biotechnology, School of Life and Applied Sciences, Yeungnam University, Gyeongsan 38541, Republic of Korea; m.singh2685@gmail.com; 3Sunforce Inc., 208-31, Gumchang-ro, Yeungcheon-si 31882, Republic of Korea; keun126@ynu.ac.kr

**Keywords:** caffeine, Alzheimer’s, nanoparticles, migraine, autoimmune diseases, metabolic disorders

## Abstract

Natural caffeine is found in many plants, including coffee beans, cacao beans, and tea leaves. Around the world, many beverages, including coffee, tea, energy drinks, and some soft drinks, have this natural caffeine compound. This paper reviewed the results of meta-studies on caffeine’s effects on chronic diseases. Of importance, many meta-studies have shown that regularly drinking caffeine or caffeinated coffee significantly reduces the risk of developing Alzheimer’s disease, epilepsy, and Parkinson’s disease. Based on the health supplements of caffeine, this review summarizes various aspects related to the application of caffeine, including its pharmacokinetics, and various functional health benefits of caffeine, such as its effects on the central nervous system. The importance of caffeine and its use in alleviating or treating cancer, diabetes, eye diseases, autoimmune diseases, and cardiovascular diseases is also discussed. Overall, consuming caffeine daily in drinks containing antioxidant and neuroprotective properties, such as coffee, prevents progressive neurodegenerative diseases, such as Alzheimer’s and Parkinson’s. Furthermore, to effectively deliver caffeine to the body, recently developed nanoformulations using caffeine, for instance, nanoparticles, liposomes, etc., are summarized along with regulatory and safety considerations for caffeine. The U.S. Department of Agriculture (USDA) and the Food and Drug Administration (FDA) recommended that healthy adults consume up to 400 mg of caffeine per day or 5~6 mg/kg body weight. Since a cup of coffee contains, on average, 100 to 150 mg of coffee, 1 to 3 cups of coffee may help prevent chronic diseases. Furthermore, this review summarizes various interesting and important areas of research on caffeine and its applications related to human health.

## 1. Introduction

Bioactive substances with a wide range of applications have long been derived from plants. Their anti-inflammatory, anti-cancer, and anti-diabetic qualities are just a few of their well-established health and wellness advantages [1,2]. Caffeine (C_8_H_10_N_4_O_2_) is a common plant-derived purine alkaloid base containing coffee and black tea. Theine or 1,3,7-trimethylpurine-2,6-dione (IUPAC) are other names for it. The structure of caffeine is shown in Figure 1. Caffeine is a primary component of many plants, including *Coffea arabica*, *Coffea canephora*, *Theobroma cacao*, *Cola acuminatam*, *Camellia sinensis*, etc. Because of its psychoactive properties, pro-arousal, pro-sympathetic, and health benefits, coffee is likely the most consumed beverage in the world [3,4]. Coffee has relatively low toxicity and contains a large amount of caffeine, a major active substance used as a food and medicine material.

Caffeine and other caffeinated brews, mostly teas and coffees, support human health by promoting the maintenance of neurological/neurodegenerative diseases, the treatment of type 2 diabetes, the prevention of cancers through its antioxidant activities, and the modulation of reactive oxygen species (ROS) production [5]. The superoxide anion (O_2_^•−^), hydroxyl radicals (^•^OH), singlet oxygen (^1^O_2_), hydrogen peroxide (H_2_O_2_), and hypochlorous acid are examples of ROS, a significant class of products during cellular metabolism [6,7]. The human body has an antioxidant defense system that can act to eliminate excess ROS, keep the antioxidant system and ROS in balance, and guarantee the body’s regular metabolism [8]. Substantial concentrations of ROS can interfere with the cellular damage repair mechanism, resulting in intracellular DNA mutations, irreversible damage to proteins and lipids, and the development of diseases [9]. This occurs when normal cells produce more ROS than the cellular antioxidant defense system can handle.

In some studies, an inverse association between dietary caffeine consumption and Parkinson’s disease (PD) has been shown in clinical research [10]. Caffeine can affect both mothers and fetuses during pregnancy [11]. Also, it was observed that steroid hormones slow caffeine metabolism in pregnancy, and caffeine slowly clears out during early infancy, possibly resulting in long-lasting effects primarily in infants’ brains [12,13]. Wikoff et al. reviewed various adverse effects of caffeine in healthy adults, pregnant women, adolescents, and children [14]. Numerous other symptoms of caffeine include elevated heart rate and respiration, anxiety, insomnia, frequent urination, dehydration, and restlessness [15]. The most common and severe side effects were tachycardia/heart palpitations and adverse effects on sleep onset in athletes who took caffeine supplements [16].

Moreover, due to the negative consequences of excessive caffeine use and its bitter taste, a method for masking this taste and designing a controlled-release delivery system for longer durations is required. The processing parameters should also be considered. For example, phenolic compounds like caffeine can be broken down by exposure to oxygen, sunlight, and the digestive system. Furthermore, because of its quick distribution and elimination rate, the peak plasma concentration is reached within 15–120 min following oral administration [17]. This illustrates the need to continuously consume caffeine throughout the day to sustain an adequate blood concentration [18]. As a result, a controlled/sustained drug release system is required, which improves patients’ compliance while reducing the frequency of dosage administration and maintaining a suitable drug therapeutic level.

To overcome the challenges of caffeine, several nano/micro-platforms have been created [19]. These include lipid-based (nanoliposomes, double emulsions, nanoemulsions, microemulsions, Pickering emulsions, niosomes, solid lipid nanoparticles, and nanostructured lipid carriers), biopolymeric (hydrogels, conjugates, nanoparticles, nanofibers, nanotubes, protein-polysaccharide cyclodextrin inclusion complexes, nano-complexes), and inorganic (silica and gold nanoparticles) nano/micro-structures [19].

Hence, this review highlights the impacts of caffeine on various health conditions and explores some emerging nanotechnology-based drug delivery systems containing caffeine.

## 2. Pharmacokinetics of Caffeine

The way the body reacts to a drug is explained by pharmacokinetics. It explores a substance’s absorption, distribution, metabolism, and excretion (ADME) [20]. Figure 2 illustrates a summary of caffeine pharmacokinetics.

Caffeine belongs to the Biopharmaceutical Classification System (BCS) Class I drugs because it is highly water-soluble and permeable. The liver metabolizes the most active components or drug(s) before they are entered into the bloodstream. But caffeine is rapidly absorbed. Hence, caffeine has minimal hepatic first-pass effects (the consequence of a drug’s rapid transformation when taken orally and resulting in reduced bioavailability) [21,22] (Figure 2). Also, caffeine is absorbed regardless of age, sex, and health, as well as the use of drug(s), alcohol, or nicotine because of minimal first-pass metabolism [23,24]. Moreover, caffeine exhibits a linear pharmacokinetic property, because a rise in caffeine dose causes a corresponding rise in blood plasma levels [25].

When caffeine is consumed, approximately 10–30% of the caffeine becomes reversibly associated with albumin and other plasma proteins [22]. Quickly and widely, the remaining caffeine molecules are dispersed throughout the various tissues and organs [26]. Following oral administration, caffeine is absorbed quickly with an absorption rate constant (*K*_01_) of approximately 0.33 min^−1^ in the small intestine. Owing to intra-individual variations and delayed gastric emptying, the time it takes to reach the highest plasma concentration ranges from 15 min to 2 h. After absorption, caffeine is dispersed throughout all tissues, body fluids, and the developing fetus. Caffeine does not accumulate in any tissue, and its distribution is 0.5 to 0.75 L/kg. The maximal plasma concentration (C_max_) level, which is equivalent to 8 to 10 mg/L, is reached by caffeine dosages of 5–8 mg/kg after around 30–75 min (t_max_). In most cases, a cup of black coffee yields a caffeine dosage of 0.4–2.5 mg/kg and a peak plasma concentration of 0.25–2 mg/L. After oral administration, it takes 45 min for the small intestine to absorb almost all of the caffeine, and it has a bioavailability of 99% with no noticeable first-pass effects [27]. Caffeine is highly bioavailable and rapidly crosses lipid membranes. After oral ingestion, peak plasma concentrations vary between 15 and 120 min, and due to rapid distribution and elimination rates, the half-life is within 3 to 5 h [28]. This means that if caffeine is not consumed consistently, blood levels will not increase [18] (Figure 2). Therefore, if needed for treatment or other health purposes, caffeine has the advantage of quickly maintaining blood levels.

Caffeine has been demonstrated to dramatically enhance cognitive function at low dosages by speeding up reaction times and visual information processing [29]. Numerous studies show that these benefits can be obtained at low dosages (starting at 0.18 mg/kg), and that the dose–response curve for cognitive responses thereafter tends to be flat [30,31]. Long-term caffeine tolerance is suggested by the fact that chronic users show reduced or even nonexistent cognitive responses [32].

Caffeine side effects may worsen with high dosages (9–13 mg/kg), but physical performance is unaffected [33]. Higher doses (~10–13 mg/kg) induced gastrointestinal side effects, mental confusion, nervousness, difficulty in focusing, and difficulty in sleeping in certain subjects [34,35], whereas lower doses (7–10 mg/kg) caused chills, nausea, flushing, palpitations, headache, and tremors [36]. When caffeine dosages were reduced to a moderate level (5–6 mg/kg), ergogenic benefits were maintained, and negative effects and physiological reactions were also reduced but not completely abolished [35]. Doses of 200 mg or higher of caffeine will cause toxicosis, which manifests as uneasiness, sleeplessness, cramping in the muscles, and times of excessive alertness [37].

Caffeine is extensively metabolized via the microsomal cytochrome P450 enzyme in the liver [20,38]. Caffeine demethylation is assisted by two of the cytochrome P450 isoenzymes, CYP1A2 and CYP2C9 [20]. Approximately 90% of caffeine metabolism is catalyzed by the CYP1A2 isoenzyme [39]. Before being eliminated from the body, practically all caffeine is processed, mostly in the liver [40]. Caffeine and its metabolites are primarily eliminated by the kidneys in urine excretion. The body excretes less than 3% of caffeine unchanged, which is a very modest amount (Figure 1). After glomerular filtration in the kidneys, reabsorption of the majority of caffeine molecules (98%) back by the renal tubules firmly into the bloodstream [41]. This illustrates how renal blood and urine flow are crucial for the removal of caffeine from the body. Caffeine undergoes oxidative N-demethylation to produce main mono-demethylated metabolites like theophylline, theobromine, and paraxanthine [42,43]. The 3-demethylation of caffeine into paraxanthine is the main first step in humans [40]. Most of the caffeine follows this pathway and 80% of it is transformed into paraxanthine, the main metabolite in humans. As seen in Figure 2, the remaining caffeine is processed to produce theophylline (~4%) and theobromine (~11%) [44].

In the below given sections, we explore recent research on the outcomes of caffeine and the use of caffeine-nanomedicine/nanoparticles on specific health conditions, such as neurodegenerative diseases, Machado-Joseph disease, neuroprotective effects, cancer, diabetes, headache and migraine management, autoimmune diseases, immunomodulation, as well as ocular, respiratory, and cardiovascular disorders.

## 3. Applications of Caffeine Function

### 3.1. Effects of Caffeine on the Central Nervous System

Caffeine is one of the common alkaloids employed alone or in combination to relieve migraines. A major barrier to efficient oral distribution of caffeine is the short elimination half-life (3 to 5 h) and the connection between its absorption from the gastrointestinal system and gastric emptying.

After oral administration, 20% of caffeine is absorbed by the stomach, and the remaining 80% is absorbed by the gastrointestinal tract [22]. Another factor affecting caffeine absorption is the oral mucosa, the mucous membrane lining the inside of the mouth [23]. Nearly 99 percent of caffeine is absorbed after oral intake, and it takes about an hour for plasma concentrations to peak [26]. In contrast, food can delay caffeine absorption as food slows down gastric emptying [26]. Also, the reduction in gastric motility can affect the caffeine absorption process and, thus, reduce its effect [45].

One of the most popular dosage forms is the transdermal drug delivery system. Compared to oral and needle-based methods, this approach offers many benefits, including self-administration, non-invasiveness, significantly reduced liver drug clearance, and increased patient compliance [46]. Additionally, by increasing the duration of the drug’s blood circulation and reducing its degradation into unwanted metabolic byproducts, it can increase the efficacy of drug(s) [47]. The Food and Drug Administration (FDA) approved many transdermal drug delivery systems of psychotropic drugs, for instance, methylphenidate, rivastigmine, and selegiline [48].

Hence, a transdermal delivery system of caffeine was developed using a three-factorial design and Design-Expert^®^ software (Version 7.0.1, Stat-Ease Inc., Minneapolis, MN) for brain delivery of caffeine for the treatment of migraine [49]. The biodistribution results from this trial showed that compared to oral suspension, transdermal 131I-loaded PNS-4 (patch) significantly improved the retention of 131I-caffeine in the blood with better brain targeting. The results of the drug delivery trial demonstrated that PNS-4 is a potentially effective transdermal drug delivery technology that can overcome the difficulties associated with oral caffeine delivery. A transdermal delivery administration reduces the amount of caffeine to maintain blood–drug concentration against oral dosage. Thus, it can be used as a targeted delivery system.

#### 3.1.1. Caffeine Effects on Amyotrophic Lateral Sclerosis and Multiple Sclerosis

Endogenous nucleoside adenosine interacts with P1 purinergic receptors to influence several physiological effects. All of them are G protein-coupled receptors, and four subtypes—A_1_, A_2A_, A_2B_, and A_3_—have been identified. G_olf_ proteins in the striatum and G_s_ proteins in the brain and the periphery are activated via stimulation of the A_2A_ receptor [50]. As a result, it controls the downstream Akt and MAPK pathways and raises cyclic AMP and PKA phosphorylation [51,52,53]. Numerous pathologies, including neurological, autoimmune, inflammatory, and malignant disorders, are impacted by the A_2A_ receptor because of its extensive distribution and the rise in its endogenous ligand in both inflammation and cancer [53]. The A_2A_ subtype of adenosine receptors has gained popularity as a target for treating several neurodegenerative diseases, for instance, neuroinflammation, including multiple sclerosis, Parkinson’s and Alzheimer’s disease, and neuropsychiatric disorders [54]. The inhibition of A_2A_ adenosine receptors (A_2A_AR) has been shown to have neuroprotective effects that counterbalance astroglial and microglial activation as well as neuroinflammatory processes.

Caffeine is well established for anti-inflammatory properties in the central nervous system and exhibits a neuroprotection function by antagonizing the excitatory A_2A_AR receptor [55,56]. The effect of caffeine consumption was investigated on amyotrophic lateral sclerosis (ALS) mortality, a neurological condition that modifies motor function and deteriorates over time until death [57,58]. Beghi and colleagues showed that patients who consumed coffee had a lower probability of acquiring ALS, raising the probability that caffeine may have a protective role in preventing ALS [58]. However, such an association was not always confirmed by subsequent studies [59,60]. In a study, caffeine was examined as a prospective treatment strategy in superoxide dismutase-1 (mSOD1G) ALS mice, and the effects of chronic caffeine administration are mostly attributed to A_2A_AR blocking [61]. Finally, they suggested that more advanced studies are required to detail the effect of caffeine on the nervous system of animal models carrying ALS disease.

Multiple neuropsychiatric co-morbidities, including depression and anxiety, are characteristics of multiple sclerosis (MS) [62]. In chronic neurodegenerative diseases like MS, the potential of A_1_AR to reduce the intensity of MS symptoms has been potentiated by the simultaneous administration of an A_1_AR agonist and a general AR antagonist like caffeine. This is likely because A_1_AR gene expression was increased by caffeine [63]. Caffeine-induced A_1A_R activation improves linked downstream signaling pathways, decreasing proinflammatory and enhancing anti-inflammatory responses, which reduce oligodendrocyte cytotoxicity and preserve the integrity of the myelin sheath and accompanying axon [63].

Consequently, long-term caffeine intake was noted with an increase in A_1_AR transcription in a guinea pig spinal cord homogenate-induced experimental autoimmune encephalomyelitis (EAE) model in rats, which was accomplished by a significant reduction in EAE symptoms [64]. Under these findings, a case-control investigation of 75 cases and 75 matched-controls of women’s daily dietary habits and MS risk revealed a negative correlation between coffee drinking and the disease risk [65]. However, another comprehensive retrospective case-control study with 2779 patients and 3960 matched controls further highlighted the benefit of coffee consumption in MS. These findings showed that drinking three or more cups of coffee daily significantly decreased the risk of developing MS [66]. Another case study found that altering A_1A_R signaling improved by using caffeine or other common medications to treat MS symptoms [67].

#### 3.1.2. Effects of Caffeine on Alzheimer’s Disease

Alzheimer’s disease (AD) is a highly prevalent type of dementia, caused by the formation of extracellular senile plaques comprised of amyloid β peptide (Aβ) and intracellular neurofibrillary tangles generated by the hyperphosphorylated Tau (τ) protein [68]. The term “amyloid beta”, also known as “Aβ”, refers to peptides with 36–43 amino acids that make up most of the amyloid plaques in the brains of AD patients. The proteolytic cleavage of the amyloid-β protein precursor (AβPP) by the β-site AβPP cleavage enzyme 1 (BACE1) and γ-secretase results in the deposition of Aβ in the brain, which is known as a significant pathogenic element in AD [69,70]. Research data suggest that the degree of amyloidogenic processing of AβPP is substantially influenced by AβPP trafficking [71,72]. When internalized (trafficked) AβPP builds up in endolysosomes, and the acidic environment boosts BACE-1 and γ-secretase activity, promoting the amyloidogenic processing of AβPP [73]. Consequently, variables that encourage the internalization of AβPP and/or disrupt endolysosome activity may increase the amyloidogenic processing of AβPP, thereby increasing the pathogenesis of AD [72]. Furthermore, the development of pathological hallmarks of AD, such as disrupted synaptic integrity, brain deposition of Aβ, and tau pathology [74], is facilitated by elevated levels of LDL (low-density lipoprotein) cholesterol, the essential lipoprotein that transports circulating cholesterol in the blood; (1) causes BBB leakage and increases brain levels of apoB [74,75]; (2) disrupts the neuronal endolysosome structure and function, another early pathological feature of sporadic AD [76].

It was found that both β-secretase (BACE1) and γ-secretase expressions are suppressed in AD transgenic mice that consume a moderate amount of caffeine (the equivalent of five cups of coffee per day), protected against developing certain cognitive impairments and have lower levels of Aβ in the hippocampus [77]. The study in AD mice was coherent with the human epidemiological literature in supporting the ability of caffeine consumption to moderate the risk of AD [78]. Hence, consuming caffeine is beneficial in preventing or delaying the development of AD before it begins, as it was observed in rats and human studies [79]. According to epidemiological research, caffeine consumption was linked with a lowered risk of developing AD in both women and men [79].

Caffeine has inhibitory activity on β- and γ-secretase, preventing Aβ aggregation and generating Aβ monomers [80,81,82]. Furthermore, the activation of adenosine receptors (A_1_R and A_2A_R) has been associated with the pathogenesis of AD [83,84], and the blockage of A_2A_R with caffeine has been shown to suppress Aβ generation [77] and protect against Aβ-induced neurotoxicity [55]. It was observed that caffeine may protect against Aβ-induced neurotoxicity in AD models by targeting the A_2A_ adenosine receptor (A_2A_AR) in cultured neurons [85].

Caffeine has also been shown to protect against Aβ-induced cognitive impairments in a transgenic mice model of AD by an A_2A_AR blockade [82,86]. Additionally, A_2A_AR knockout mice decreased their locomotor activity following caffeine administration and showed that caffeine primarily produced stimulant effects by inhibiting the A_2A_AR [87]. Furthermore, by inhibiting the A_3A_AR-mediated internalization of the amyloid precursor, caffeine inhibited the amyloid formation and showed a protective effect [72]. Caffeine prevents A_3_AR-mediated internalization of AβPP from having protective effects against the amyloidogenic processing of AβPP by regulating neuronal internalization of LDL cholesterol [72]. Also, it has been confirmed that coffee intake with AD biomarkers in the human brains of 411 MCI patients who were not yet diagnosed with dementia. The results suggested that drinking two or more cups of coffee per day over a lifetime was substantially associated with lower plaques composed of Aβ compared to not drinking Aβ [88].

#### 3.1.3. Effects of Caffeine on Parkinson’s Disease

Parkinson’s disease (PD) is a chronic, progressive neurological disease that causes bradykinesia, slow movements, postural instability, and rigidity [89]. Caffeine is considered a putative drug for the treatment of PD [90,91]. Some potential ways to delay the progression of Parkinson’s disease using the neuroprotective drug caffeine have been proposed [92]. These include blocking the A_2A_AR-mediated tonic suppression of the dopamine D2 receptor (D2R) signaling in the striatopallidal pathway, which limits the Lewy Body (LB) formation, thereby creating antioxidant activity [90,93,94]. Convergent epidemiological and experimental studies suggested that caffeine may provide neuroprotection against the underlying degeneration of dopaminergic neurons and may influence the onset and course of PD [79].

The glutamate ionotropic receptor NMDA type subunit 2A (GRIN2A) gene may have a single nucleotide polymorphism (SNP) that causes the dopaminergic neuronal degeneration associated with PD [95]. According to two studies, the relationship between caffeine and the risk of PD is significantly influenced by the T-allele of the GRIN2A single nucleotide polymorphism (SNP) rs4998386 [96,97]. According to Hamza et al., heavy coffee drinkers who carried the GRIN2A T allele were 59% less likely to develop Parkinson’s disease than light coffee drinkers [96]. A study by Simon et al. of 1741 patients with early PD revealed a complex relationship between the rate of PD progression, caffeine, creatine, and *GRIN2A* genotype [98]. They concluded that a *GRIN2A T* allele is linked to a markedly higher rate of PD progression in individuals taking creatine and consuming large amounts of caffeine. While the placebo group did not have evidence of a relationship between the *GRIN2A* genotype and the rate of disease advancement [98].

Another study by the Postuma group (2017) of 60 patients receiving caffeine and 61 receiving a placebo, collected from a multicenter US–Canada study, found that caffeine did not provide long-term symptomatic benefits for parkinsonism in motor PD, but only with temporary altering effects [99].

Gelber and colleagues (2011) investigated the neurological illnesses that cause cognitive impairment [100]. Their study included 3734 Japanese American males ranging in age from 71 to 93 years old. They observed that when caffeine intake was high, there were fewer cortical LBs found at autopsy. This can be explained by the fact that caffeine binds to the protein α-synuclein, modifying the conformation of its fibrillary aggregates and preventing them from aggregating [101].

A meta-analysis study including a cohort of 184,024 people and 351 cases of incident PD found that pre-diagnostic caffeine, along with its primary metabolites paraxanthine and theophylline, were associated with a reduced risk of PD. The odds ratios (ORs) were observed 0.80 (95% CI: 0.67–0.95, *p* = 0.009) for caffeine, 0.82 (95% CI: 0.69–0.96, *p* = 0.015) for paraxanthine, and 0.78 (95% CI: 0.65–0.93, *p* = 0.005) for theophylline. Years before its diagnosis, coffee’s neuroprotective effects on PD have been linked to caffeine and its metabolites [102].

The pathophysiology and development of PD are significantly influenced by redox dysfunctions and neuro-oxidative stress. Moreover, the antioxidant properties of caffeine were shown by different in vivo and in vitro studies [66,103,104], suggesting it has neuroprotective effects by reducing the mitochondrial stress in PD. The increased activity of the enzymes glutathione reductase (GSH) and superoxide dismutase (SOD), which prevent lipid peroxidation, after caffeine consumption, supports the antioxidant benefits of caffeine [105]. According to in vitro and in vivo research, caffeine increases SOD2 activity by preventing acetylation-induced deactivation. The reason for this is its affinity for sirtuin 3 (SIRT3), a mitochondrial nicotinamide adenine dinucleotide (NAD+)-dependent acetylase, which is involved in the deacetylation of two SOD lysine residues (lysine 68 and 122) [106]. Caffeine’s affinity for the substrate is enhanced by its binding to SIRT3, which also boosts the antioxidant capability of the enzyme [107].

On the other hand, oxidative stress is brought on by an excess of ROS or a weakened antioxidant system, which results in neurodegeneration. The most prevalent thiol-reducing agent and a key antioxidant in the central nervous system is GSH. Under oxidative stress circumstances, glutathione (GSH) and the reduced/oxidized glutathione (GSH/GSSG) ratio are decreased, which may result in increased oxidative toxicity [108]. It was observed in vivo that an intraperitoneal injection of caffeine into male C57BL/6 mice significantly increased the total GSH levels in the hippocampus of the brain [109].

To summarize, in Table 1, we presented important studies on the relationship between coffee or caffeine consumption and the risk of dementia, AD, and PD. The meta-studies examining more than 10,000 samples listed in Table 1 found that drinking caffeine regularly significantly reduced the risk of developing PD in adults (Table 1). Therefore, these results indicated that caffeine intake can be considerably correlated with a lower risk of PD development.

#### 3.1.4. Effect of Caffeine on Epilepsy

Approximately 1% of the population suffered from epilepsy, a serious medical condition marked by recurring seizures. Various studies revealed a connection between epileptic activity and inflammation of the central nervous system [120]. It was demonstrated that caffeine could protect neuronal cells in rats against epileptic injury, whereas acute high-dose caffeine had negative effects in animal models of epilepsy [121]. It has also been observed that giving caffeine to genetically absent epileptic WAG/Rij mice decreases the quantity of SWD (spike-wave discharge) at 30 and 60 min, but increases brain tissue levels of NFkB and IL-6, as well as blood levels of TNF-α and IL-1β [122]. These findings demonstrate that low-dose acute caffeine treatment affects epilepsy by inducing systemic inflammation and suppressing inflammatory responses in thalamic tissue [122]. The study concluded that the adenosine A_1_AR and A_2_AR receptors’ antagonistic properties of caffeine resulted in a decrease in the number of SWDs when given acutely at low doses in the GAERS (Genetic Absence Epilepsy Rats from Strasbourg) rat strain [123].

#### 3.1.5. Effect of Caffeine on Huntington’s Disease

Long CAG repeats are responsible for Huntington’s disease (HD), a genetic neurodegenerative disorder characterized by abnormalities of the motor, cognitive, and mental domains [124]. According to one epidemiological investigation, caffeine usage levels of more than 190 mg/d were related to an increased risk of developing HD [124]. Caffeine has already been shown in animal models to improve motor function. However, the contradiction between epidemiological and animal research needs to be investigated further to check the effects of caffeine on HD because it was not possible to deduce a relationship or lifestyle recommendations from the data [124,125]. Another study evaluated that higher caffeine intake may be linked to lower HD mortality, according to a nationwide study [126]. The findings could potentially assist medical professionals in recommending lifestyle modifications that can reduce mortality in hereditary illnesses like HD. Results must be confirmed by other research, though, including studies with bigger HD sample populations and longer monitoring times [126]. So, it can be presumed that caffeine intake in the cure of HD is not fully confirmed, so more studies are needed to confirm the correlation between caffeine intake and HD.

#### 3.1.6. Effect of Caffeine on Machado-Joseph Disease

Machado-Joseph disease (MJD) is also caused by an increase in the frequency of repetition of the CAG codon. It was recognized by its adult-onset age and early death linked to an unstable CAG stretch expansion in the MJD1 gene, which produced the related ataxin-3 protein’s polyQ (polyglutamine) repeat [127]. One study found that long-term coffee use and genetic A2AAR deletion reduced neuronal mortality, neuronal dysfunction, reactive gliosis, progressive degeneration, and sequestered noxious ataxin-3, and prevented ataxin-3-induced synaptotoxicity [128]. An epidemiological study is required to conclude the long-term impact of caffeine usage on the onset and/or progression of MJD. Further research utilizing animal models is also necessary to fully understand the mechanisms that link caffeine to MJD.

### 3.2. Other Neurological Applications

#### 3.2.1. Caffeine and Heavy Metals

Heavy metals, which have been utilized by humans for many years, are extremely detrimental to both the environment and the human population’s health. Cadmium (Cd), mercury (Hg), zinc (Zn), nickel (Ni), chromium (Cr), lead (Pb), and copper (Cu) are the seven most prevalent heavy metals, and they are all harmful to both humans and the environment [129,130,131]. Amidst these, Cd is a ubiquitous, non-biodegradable environmental contaminant with several harmful impacts on people [132]. Major human organs like the brain, liver, kidneys, heart, lungs, and bones collect Cd, which damages both their structural and functional integrity. Its half-life is around 7 to 30 years [133,134,135]. The brain is the primary organ affected by Cd-induced neurotoxicity because it increases the blood–brain barrier (BBB) permeability, causes neuroinflammation due to oxidative stress, and behavioral impairments [136,137]. This leads to neurodegenerative disorders like Parkinson’s disease and Alzheimer’s [138,139]. Most notably, male C57BL/6N mice, as well as HT-22 and BV-2 cell lines, were used to test the neuroprotective effects of caffeine (30 mg/kg) against Cd (5 mg/kg)-induced oxidative stress-mediated neuroinflammation, neuronal death, and cognitive impairments. Overall, the results showed that caffeine protected against memory loss, neuroinflammation, and oxidative stress-mediated neurotoxicity caused by Cd. This revealed that caffeine might also have antioxidant and neuroprotective properties that might help to prevent Cd-induced neurodegeneration [132]. Furthermore, an investigation was conducted into the degree of MeHg and HgCl_2_-mediated cytotoxicity towards SH-SY5Y human dopaminergic neurons. Caffeine and interferon (IFN-) can assist in decreasing it by controlling glutamate-mediated signaling. The study found that caffeine protected against MeHg-induced toxicity via glutamate transmission, but this protective effect was only partially reversed when caffeine was co-stimulated with glutamine and IFN-γ [140].

#### 3.2.2. Caffeine and Management of Headaches, Migraines, and Associated Chronic Pain, Including Neuropathic Pain

According to Nouri-Majd et al., 3362 participants (Iran; age 36.2 ± 7.8 years) in a crossover design for 14 weeks showed no significant correlation between caffeine intake and odds of symptoms of anxiety (OR 0.90; 95% CI (0.67, 1.20)), depression (OR 0.94; 95% CI (0.75, 1.16)), and psychological distress (OR 1.13; 95% CI (0.89, 1.42)). The experiment on caffeine intake and the symptoms of adult psychological disorders showed that weekly or higher coffee intake was linked with a considerably decreased risk of having anxiety symptoms when compared to no coffee consumption [141]. The reported clinical study indicated that coffee 2–3 cups intake a day could be advised as part of a healthy lifestyle to enhance mental wellness [142]. Remarkably, a study involving 40 healthy participants involved either a coffee group (coffee group) or a juice group (juice group) that received modest levels of caffeine for four weeks. In contrast to the juice group, the coffee group’s effects on PPT and PPT decreased after 4 weeks, while EA’s effects on NRS scores and the RIII reflex did not change. In the second week, there was no noticeable change in these signals. In healthy volunteers, the effects of EA on PPT and PPT were lessened by moderate coffee use [143]. In addition, in another clinical study with 422,586 participants, several meta-analyses were conducted on the connection between caffeine consumption and depressive symptoms, as well as the impact of caffeine on anxiety episodes in patients suffering from panic attacks [144]. Caffeine consumption was shown to be substantially linked in multivariate regression models adjusted for covariates with enhanced heart pain threshold (β = 0.296) and pressure pain threshold (β = 0.277). Those who used caffeine regularly showed reduced sensitivity to painful stimuli [145]. A recent report showed that caffeine triggers panic attacks in a significant number of Parkinson’s disease (PD) patients, and this population is markedly different from healthy people at levels roughly equivalent to five cups of coffee [144]. Moreover, caffeine also increases anxiety in healthy adults and Parkinson’s disease patients. Although, it is unclear how precisely panic attacks and anxiety brought on by caffeine are connected [146]. A recent study also proved that caffeinated coffee intake is moderate (1–2 cups/day), high (≥3 cups/day), and no-to-low (<1 cup/day). Among the 3030 respondents to the survey, 170 (5.6%) and 1768 (58.3%) were found to have headaches other than migraines. Caffeinated coffee intake and psychiatric comorbidities were found to be correlated; increased caffeine intake was linked to decreased stress and depression levels [4].

### 3.3. Effect of Caffeine on Cancer

Caffeine and other methyl xanthines have anticancer properties as shown in numerous studies. Inhibiting kinases, inducing DNA repair aspects in cancer cells, preventing the development of cancer, modifying the immune system, reducing drug toxicity, and boosting the therapeutic properties of drug(s) are just a few of the anticancer properties of caffeine and its derivatives.

Kumar et al. prepared a novel amphiphilic biconjugate of caffeine, which can self-assemble into nanospheres in water [147]. Nanospheres were easily encapsulated by the anticancer drug doxorubicin, stable in serum, and in a reducing environment of dithiothreitol and glutathione, they could be easily disintegrated. They showed the efficient transport of drugs into doxorubicin-resistant cervical cancer (HeLa) cells. in vitro, results such as cell viability and hemocompatibility suggested that the nanospheres were safe and efficient.

Breast adenocarcinoma (MCF-7) cells treated with a combination of sub-cytotoxic quantities of an organometallic platinum(II) terpyridine complexes and an N-heterocyclic carbene ligand generated from caffeine showed a substantial anticancer impact [148].

A lipid-based nano-carrier was used to load imiquimod (R837) and caffeine as a nano-immunomodulator for combination therapy with radiation. The nano-carrier has the following properties: size, polydispersity index (PdI), and zeta potential—154, 0.147, and −23.4—respectively. Compared to the IL-4 (interleukin 4)-treated control, the expression of iNOS (nitric oxide synthase), which causes tumor cell death, significantly increased after the nano-immunomodulator therapy, rising from 19.4% to 96.1% (76.7%) [149].

In other nanoformulations, considerable anticancer action was demonstrated in magnetic hyperthermia by N-heterocyclic carbene silver complex anchored on Fe_3_O_4_ NPs functionalized by caffeine metabolite [150].

Colorectal cancer is the leading cause of cancer-related fatalities, and it accounts for the second-highest incidence in women and the third-highest in men [151]. The correlation between coffee drinking and cancer risk has already been examined in certain epidemiological studies [152,153]. According to Sun, F. et al., caffeine exerts anticancer effects by inducing apoptosis and suppressing cell growth via the FoxO1 signaling pathway [154]. In a different cohort study, 6989 men from the Moli-sani (biggest cohort study) were examined to determine the impact of Italian-style coffee consumption on prostate cancer risk and to examine the possible antiproliferative and antimetastatic activity of caffeine on tumor cell lines. It was discovered that caffeine appeared to have antimetastatic and antiproliferative properties on two distinct cell lines used to study prostate cancer (PC-3 and DU145) [155]. The relative risks (RRs) of HCC, based on the consumption of caffeinated and decaffeinated coffee, were determined through a random-effects dose–response meta-analysis utilizing a thorough analysis of 18 cohorts, comprising 2,272,642 participants and 2905 cases, and 8 case-control studies, encompassing 1825 cases and 4652 controls. A decreased risk of HCC has been associated with increased use of caffeinated coffee and, to a lesser extent, decaffeinated coffee, particularly in individuals with pre-existing liver disease [156]. In observational studies published in February 2016, a literature review and meta-analysis on the correlation between tea and coffee drinking and the risk of NMSC (nonmelanoma skin cancer) were investigated. A total of 37,627 NMSC cases from 13 articles were included in the study. The findings indicated that caffeine intake was negatively correlated with the incidence of NMSC. They concluded that coffee consumption can have a modest protective effect against BCC (basal cell carcinoma) formation, possibly due to the biological effects of caffeine [157]. Another meta-analysis investigation of a total of seven studies that were qualified for meta-analysis, including 1,418,779 participants and 9211 incidences of melanoma. According to this study, melanoma risk and overall coffee intake are significantly correlated. A daily increase in one cup of coffee was linked to a 3% decrease in the chance of developing melanoma. This important finding only applied to caffeinated coffee but was not found to decaffeinated coffee [158]. Similarly, significant prospective cohort research on Canadian women (39,532 female participants) found that caffeine and/or coffee use may reduce the threat of endometrial cancer but, on the other hand, may raise the threat of breast cancer in premenopausal or normal-weight women [159]. On the contrary, according to another case-control study from the Nurses’ Health Study and Nurses’ Health Study II, caffeine and regular coffee were found to have inverse relationships with breast density which lowers the chances of developing breast cancer [160]. This was consistent with prior meta-analysis research that included 26 studies (16 cohorts and 10 case-control studies) on coffee intake, which included 49,497 instances of breast cancer. The combined RR indicated that the risk of breast cancer was rather significantly influenced by the highest coffee consumption (RR = 0.96; 95% CI 0.93–1.00), low-to-moderate coffee consumption (RR = 0.99; 95% CI 0.95–1.04), or an increase in coffee consumption of two cups per day (RR = 0.98; 95% CI 0.97–1.00) [161].

A decreased incidence of inflammatory bowel disease (IBD) was linked to higher levels of caffeine, with an odds ratio (OR) of 0.78 (95% confidence interval [CI]: 0.66, 0.91; PFDR = 0.004). Moreover, this tendency was noted with ORs of 0.79 (95% CI: 0.66, 0.94; PFDR = 0.014) and 0.78 (95% CI: 0.62, 0.98; PFDR = 0.032) for ulcerative colitis (UC) and Crohn’s disease (CD). Coffee has a lower risk of Crohn’s disease (CD) and ulcerative colitis (UC), two kinds of inflammatory bowel illness [162]. The multiethnic cohort study (MEC) (including 167,720 participants of African Americans, Native Hawaiians, Japanese Americans, Latinos, and White Americans) was established in Hawaii and Los Angeles and observed low threat of thyroid, liver, ovarian, and endometrial cancers, as well as melanoma with increased coffee consumption. Only overweight women were shown to have a reduced probability of endometrial cancer as a result of coffee drinking [163]. The study on human GBM (glioblastoma multiforme) cell line U87-MG found that caffeine increased the TMZ’s (Temozolomide, a first-line chemotherapeutic agent used to treat the glioblastoma multiforme) chemo-efficacy by inhibiting the ATM/p53/p21 pathway and the promoting mitotic catastrophe [164]. According to experimental research conducted on subjects of the Singapore Chinese Health Survey, a population-based prospective cohort of 63,257 men and women (n = 27,959 and n = 35,298, respectively), a high caffeine intake may reduce the risk of NMSC (nonmelanoma skin cancer). Coffee consumption was specifically linked to lower risks of SCC (squamous cell carcinoma) and BCC (basal cell carcinoma), whereas black tea consumption was linked to lower risks of SCC [165]. Moreover, Ohata et al. demonstrated that caffeine, as well as a specific A_2_AR antagonist, ZM241385, increased the ability of tumor antigen-specific CD8^+^ T lymphocytes to destroy implanted tumor cells [166]. Hence, the use of caffeine as a pharmacologic A_2_AR antagonist to promote anti-tumor immune response during tumor initiation was suggested as therapeutic potential against cancer development [167]. Due to the blockage of A_2B_AR, caffeine has recently been demonstrated to have anticancer effects in animal research [168].

Moreover, the association between coffee drinking and the incidence of nonmelanoma skin cancer was ascertained using a thorough review and meta-analysis of cohort and case-control studies. This meta-analysis suggests that caffeinated coffee may offer dose-dependent benefits for chemoprevention against basal cell cancer [169]. Table 2 summarizes many studies of the relationship between coffee drinking or caffeine consumption and the risk of different types of cancer.

### 3.4. Effects of Caffeine on Diabetes

Several meta-analysis studies have confirmed that people with type 2 diabetes respond to caffeine differently as shown in Table 3. It can increase insulin and blood sugar levels in people with the illness. This may have resulted from enhanced glucose regulation brought about by a rise in cyclic adenosine monophosphate (cAMP) levels in pancreatic beta-cells. One particular study determined and used the relationships that exist between the human body’s levels of sorbitol, caffeine, insulin, and blood glucose for possible therapeutic uses in treating diabetic neuropathy [210]. In a meta-analysis, Ding et al. found that drinking coffee, both caffeinated and decaffeinated, reduced the risk of developing diabetes. However, no noticeable difference was found between the two types of coffee (pooled RR of T2D for 1 cup/day increase = 0.91, 95% CI = 0.89–0.94, and 0.94, 95% CI = 0.91–0.98, respectively, *p* for difference = 0.17) [211]. A marginally (*p* = 0.053) lower risk of pre-diabetes was associated with high dietary caffeine intake (≥152 vs. <65 mg/d) (HR = 0.45, 95% CI = 0.19–1.00). Regular diets containing caffeine may be beneficial in preventing pre-diabetes. Additionally, our results suggested that an increased coffee and caffeine intake might alter insulin resistance and insulin levels [212]. The risk of type 2 diabetes was found to be negatively connected with five cups (177 mL) of coffee per day for eight weeks, as per previous investigations including 45 healthy overweight female volunteers [213]. An interesting study found that over 24 weeks, ingesting 4 cups of caffeinated coffee per day did not significantly alter biological mediators of insulin resistance or insulin sensitivity, but it was associated with a slight reduction in FM and a decrease in urine creatinine concentrations [214].

Coffee use regularly may decrease the consequence of type 2 diabetes by avoiding the impairment of liver and beta cell function under chronic metabolic stress before the onset of type 2 diabetes (i.e., overt diabetes) [222].

### 3.5. Effects of Caffeine on Autoimmune Diseases

The cohort study found a positive correlation between the frequency of rheumatoid factor positivity and the number of cups of coffee drunk daily, suggesting that coffee may have an effect on autoimmune diseases, such as rheumatoid arthritis (RA) [223]. Caffeinated coffee intake correlated with RA risk (caffeinated coffee: pooled effect size 1.30, 95% CI: 1.09 to 1.54, I2 = 0%) and confirmed that it was associated with increased RA risk [224]. The results of clinical trials involving two cohorts of participants, 15,551 women from the Nurses’ Health Study and 7397 men from the Health Professionals Follow-Up Study, revealed that the association between the increased risk of RA and caffeinated coffee consumption (≥4 cups/day) is related to the impact of coffee on the levels of inflammatory markers, including c-peptide, estrone, total estradiol, free estradiol, leptin, CRP, IL-6, sTNF-2, and total adiponectin [225]. According to Lee et al., there was a strong correlation between the intake of decaffeinated coffee and the risk of RA in a case-control study. However, the meta-analyses of cohort studies did not yield statistically significant results [226]. Regular and decaffeinated coffee consumption was found to have a significant positive connection with the risk of RA in a recent meta-analysis [227]. Furthermore, a different cohort study, including 76,853 women, examined the moderating impact of regular coffee drinking on the risk of RA [228].

### 3.6. Immunomodulation

#### 3.6.1. Effects of Caffeine on Ocular

In a meta-analysis with 711 myopic children (Denmark) study, caffeine metabolite 7-methylxanthine (0–1200 mg per day) was coupled with decreased myopia progression and axial elongation [229,230]. One Spanish cohort with 22 university students (21 years old) reported that caffeine (4 mg/kg) consumption was associated with a reduced variability of accommodative responses and induced pupil dilations [230]. In Germany, seventeen healthy volunteers (29.6 ± 3.73 aged), who orally ingested 200 mg of caffeine, saw a significant reduction in artery and venule width of 4.9% and 6.7%, respectively [231]. In contrast, another Eye Research Australia cohort study reported that caffeine 300 mg (3 cups) is linked with an acute vasoconstrictive consequence on retinal venules and arterioles in healthy subjects [232]. Eighteen participants, six of whom were male and aged 24.3 ± 3.1, participated in the Law et al. study. The findings indicated that individuals with high myopia who took 200 mg of caffeine had decreased sub-foveal choroidal thickness and restricted retinal vasculature of the superficial capillary plexus [233]. It has also been demonstrated that one-hour following oral coffee consumption, there is a considerable decrease in peripapillary and macular vessel densities, using optical coherence tomography angiography and markers of macular flow area [234]. High caffeine consumption was associated with a low probability of diagnosing dry eye disease (DED), according to a prospective cohort study by Magno et al. involving 85,302 participants, of whom 59% were female [235]. A meta-analysis performed on 121,374 participants (39–73 years old) reported by Kim et al., assessed that caffeine consumption was associated with a 0.10 mmHg lower intraocular pressure (IOP) [236].

#### 3.6.2. Effects of Caffeine on Respiratory

Caffeine (10 and 20 mg/kg daily) was found to reduce the nasal continuous positive airway pressure in neonates with respiratory distress syndrome (RDS), according to a randomized controlled experiment, including 90 neonates) [237]. Doyle et al. found consistent results with 142 children in their study, and it seems that caffeine treatment at the newborn stage improves respiratory health during the newborn phase, which in turn improves expiratory flow rates in mid-childhood [238].

### 3.7. Effects of Caffeine on Cardiovascular Disorders 

Globally, cardiovascular disease (CVD) is on the rise and can account for most of the public health burden coupled with mental health issues, particularly in a world where climate change is having an impact. There is a strong correlation between behavior, psychological mechanisms, and CVD. A recent meta-analysis review showed that coffee drinking dramatically raised the risk of CHD in the follow-up of 20 years or more (RR 1.16, 95% CI 1.06 to 1.27, n = 4) [239]. A randomized meta-analysis of adult participants ≥ 18 years old tested systolic (SBP) and diastolic (DSP). A pooled analysis of 11 effect sizes from 8 trials showed that there was a significant rise in both SBP (WMD: 1.94 mmHg; 95%CI: 0.52, 3.35; *p* = 0.007) and DBP (WMD: 1.66 mmHg; 95% CI: 0.75, 2.57; *p* = 0.000) with caffeine consumption [240]. In the Physicians Health study, which involved 20,433 middle-aged and older men, there was no association found between the consumption of caffeine in food and the risk of heart failure (HF). The multivariable-adjusted HR (95% CI) was 1.07 (0.87–1.31), 0.95 (0.77–1.18), 1.06 (0.86–1.31), and 1.15 (0.92–1.44) for the successive quintiles of caffeine consumption (*p* for linear trend—0.34). There is no association between the prevalence of heart failure and coffee consumption or dietary caffeine intake among male physicians in the United States [241]. A related study among adults with 362,571 Biobank participants indicated that long-term coffee (>6 cups/day) may result in an increased lipid profile (LDL-C, total cholesterol, and apolipoprotein in plasma, raising the CVD in affected individuals [242]. In another study, 2278 US volunteers (18–80 years old) testified that caffeine metabolites were liable for decreasing the risk of hypertension [243].

According to Feng et al.’s study, which included 23,878 people older than 20, the all-cause mortality was considerably lower in subjects who drank 100–200 mg of caffeine daily (hazard ratio: 0.78; 95% CI: 0.67–0.91) than those who consumed more than 200 mg (hazard ratio: 0.68; 95% CI: 0.60–0.78). Lower CVD mortality was linked to higher caffeine intake (>100 mg/day) [244]. Twelve people, aged 19 to 39, participated in the study. When caffeine (200 mg, given at 12 h intervals) is administered while sleep-deprived, HR is decreased and HF-HRV is raised. There was a nonlinear concentration effect. There is no discernible relationship between coffee consumption and sleep deprivation [245]. The CVD group used 209.99 ± 196.76 mg of caffeine daily, according to an investigation confirmed that blood markers associated with cardiovascular illnesses may be partially impacted by caffeine use [246].

### 3.8. Cosmeceutical Application of Caffeine

Cosmeceutical nano gels based on caffeinated hyalurosome nanoformulations can overcome the drawbacks of caffeine for cosmetic use, such as limited permeability and skin deposition. This formulation, comprising phospholipid vesicles and a hyaluronan polymer with a mean size of 210.10 nm, zeta potential of −31.30 mv, and encapsulation efficiency of 84.60%, demonstrated 4.89-fold higher skin accumulation in dorsal rat skin after 24 h of incubation when compared to a control gel formulation (Carbopol-940 loaded with caffeine) [247].

Caffeine-loaded semisolid nanostructured lipid carriers (CAF-NLCs) were synthesized by high-speed homogenization and ultrasonication for use as a topical cellulite therapy [248]. The particle size, polydispersibility index, zeta potential, viscosity, and CAF content values were found to be 318.8 nm, 0.253, −41.1 mV, 18.0 Pa·s, and 97.57%, respectively for optimized CAF-NLC3. When compared to ordinary CAF gel (CAF-P) and commercial CAF gel (CAF-M), a single topical application of optimized CAF-NLC3 to the shaved abdomen skins of Wistar rats showed higher skin retention of CAF by 2-fold and 1.4-fold after 4 h, respectively. Additionally, by upregulating the expression of PPAR-γ and UCP1, and promoting the brown adipose tissue thermogenesis by upregulating the levels of extracellular matrix components (collagen 1, elastin, and hyaluronic acid), optimized CAF-NLC3 demonstrated higher anti-cellulite action when compared to CAF-P and CAF-M. Thus, adding CAF to a semisolid NLC formulation would be a cosmetic strategy that shows promise for treating cellulite topically.

Caffeine-loaded nanostructured lipid carriers derived from coffee silverskin (NLC-CS), were developed and evaluated via a double emulsion technique [249]. NLC-CS, with a zeta potential of around −30 mV and a low polydispersity index of less than 0.25, were in the nanosized range (200 nm) and showed storage stability for up to 180 days at 25 °C/65% with relative humidity (RH) and 40 °C/75% RH. in vitro skin penetration showed its capacity to cross the skin barrier when encapsulated in nanoparticles and suggested plausible topical therapy for cellulitis.

## 4. The Need for Bioengineering Caffeine for Health Benefits

### Nanotechnology-Based Delivery Strategies

Caffeine consumed through coffee drinks has limitations in medicinal efficacy. Currently, most caffeine is consumed through beverages, especially coffee, tea, and medications, so it is quickly absorbed and distributed to all tissues [250]. Because we have recognized in this review that caffeine has various health benefits, it is often necessary to administer caffeine to targeted tissues. However, caffeine only lasts for three to five hours [49]. Furthermore, orally consuming large amounts of caffeine can result in gastrointestinal issues [251] and its widespread use may cause adverse side effects, such as nervous system stimulation. Additionally, since caffeine is easily soluble in water compounds, a controlled/sustained drug release method is required to deliver it. The multifaceted discipline of nanotechnology by biotechnology and pharmaceuticals allows for the production of new delivery systems with special features and the manipulation of materials at the nanoscale (1 to 100 nm) [252].

The nanotechnology technique not only reduces the frequency of dosage administration and maintains the therapeutic level of the drug but also improves patient compliance. To properly maintain a patient’s medication with a gradual rise in the blood level without any burst effect, a sustained drug delivery system will be a preferable choice. The sustained drug release via nanoparticulate drug delivery systems can be achieved through a variety of methods, including liposomes, solid lipid nanoparticles [253,254], and nanospheres [255].

Drugs, which are chemical substances, can cause unexpected toxicity when exposed directly to cells. Drugs can be administered to cells by using nano-vesicular systems without causing any significant negative effects. It has recently been discovered by researchers that non-ionic vesicles made from Span 80 (sorbitan monooleate) can be useful for caffeine delivery [256]. Using this nano-delivery method, caffeine-potentiated chemotherapy showed notable anti-tumor effects with insignificantly harmful side effects. Despite being tested on an animal model, results suggested that osteosarcoma therapy may benefit from its application in clinical settings. Moreover, caffeine can also be administered via nanoemulsions, which is another drug delivery technique [257]. One of the most promising techniques for transdermal delivery is the use of nanoemulsions, which can improve the drug’s skin penetration and bioavailability. Caffeine has low skin retention and ineffectively penetrates the lipophilic barrier of the outermost skin layer due to its high hydrophilicity. Therefore, water in oil nanoemulsion formulations of caffeine can be employed as carriers for effective transdermal administration.

A study also demonstrated the potential of ideal microemulsion formulations as topical nano-carriers for the administration of caffeine [258]. The findings indicated that the use of microemulsions could improve skin retention and the therapeutic effects of caffeine on skin tumors. Consequently, data suggest that microemulsions could be a potentially effective topical drug(s) delivery system for caffeine. Research has indicated that caffeine can offer protection against skin cancer caused by UVB exposure.

Drug distribution can also be facilitated using nanoparticles. Many cosmetic treatments, such as lotions designed as anti-cellulite, contain caffeine or its derivatives as their primary constituent. Caffeine nanoparticles (NIPAM-co-AAc) copolymerized with acrylic acid have shown to be promising carriers for transdermal administration of caffeine for dermatological and cosmetic applications [259]. In the new technology, nanoparticles containing caffeine have been shown to penetrate the skin barrier better than regular caffeine-containing creams. Examples of recent caffeine drug delivery biotechnology are summarized in Table 4.

## 5. Caffeine Use Regulations and Safety Considerations

The amount of caffeine consumed varies among various beverage varieties and demographic groupings. Coffee is extensively consumed and usually contains more caffeine than most other beverages. Hence, caffeine has been a part of the human diet for many millennia and, if taken in moderation, is generally regarded as safe (GRAS). However, the rise of energy drinks in the market more recently caused a shift in the beverage industry. There are concerns about the potential health effects of excessive caffeine consumption because energy drinks are a new and much higher-level dietary source of added caffeine. Excessive caffeine consumption can result in anxiety, headaches, nausea, restlessness, and an increased risk of hypertension and cardiovascular disease, whereas moderate caffeine consumption can enhance mental alertness, focus, tiredness, and physical performance [272,273,274].

The U.S. Department of Agriculture (USDA) and the Food and Drug Administration (FDA) recommend that healthy adults can consume up to 400 mg of caffeine per day, or 5~6 mg/kg body weight for a 65 kg individual, which does not substantially increase the risk of these diseases in healthy persons [274,275]. Although there are no particular guidelines for the adolescent demographic, pre-adolescents are encouraged to keep their daily caffeine consumption to no more than 2.5 mg/kg body weight [275]. The U.S. Code of Federal Regulations lists caffeine as GRAS for cola-type beverages, allowing 200 ppm or 0.02%, and a total of 71 mg for a 12 oz (354 mL) soft drink.

The FDA recommended that manufacturers list the caffeine concentration of their products on labels and apply its GRAS standard to energy drinks and other beverages that include additional caffeine. Since a cup of 354 mL coffee contains, on average, 150 mg of coffee, 2 to 3 cups of coffee may be considered appropriate to help the preventive effect of chronic diseases. The research and medical sectors generally agree that children should not consume any caffeinated energy drinks due to the negative and potentially dangerous effects of caffeine on development. Considering the limitations in the intake of caffeinated beverages, biotechnologies to deliver caffeine locally to the area or tissue in need of treatment are becoming more important.

## 6. Conclusions

Caffeine is naturally or synthetically synthesized and extensively consumed around the world as a medicine or a beverage form, whereas the quantity taken differs from country to country. After oral administration, 20% of caffeine is absorbed by the stomach, and the remaining 80% is absorbed by the gastrointestinal tract. Both animal and human experiments indicate that caffeine is safe if it is consumed in moderate quantities. The regulatory agencies in the U.S. recommended that adults consume up to 400 mg of caffeine per day or about 5–6 mg per kg of body weight. Caffeine is readily absorbed regardless of age, sex, and health, as well as the use of drug(s), alcohol, or nicotine, due to minimal hepatic first-pass metabolism.

By helping to maintain neurological and neurodegenerative diseases, treat type 2 diabetes, prevent cancer through antioxidant properties, and regulate the production of reactive oxygen species (ROS), caffeine and other caffeinated beverages like tea and coffee support human health. Numerous neurodegenerative diseases, such as neuroinflammation, which includes multiple sclerosis, Parkinson’s and Alzheimer’s disease, and neuropsychiatric disorders, may be influenced by caffeine’s potential to antagonize the A_2A_ subtype of adenosine receptors (A_2A_AR). The majority of effects of long-term caffeine intake are attributed to A_2A_AR blocking. According to some studies, there may even be a negative correlation between Parkinson’s disease and dietary caffeine intake. Additionally, caffeine may have neuroprotective and antioxidant qualities that help stop Cd-induced neurotoxicity.

Furthermore, caffeine and its derivatives have anti-cancer effects that include inhibiting kinases, causing DNA repair properties in cancer cells, delaying cancer formation, altering the immune system, lowering drug toxicity, and improving the therapeutic properties of drug(s). Successively, it was found that regular coffee consumption can prevent damage to liver and beta cell function under chronic metabolic stress prior to the onset of type 2 diabetes, possibly reducing the consequences of the disease.

Due to its effect on inflammatory markers such as c-peptide, estrone, total estradiol, free estradiol, leptin, CRP, IL-6, sTNF-2, and total adiponectin, caffeinated coffee consumption was linked to an increased risk of rheumatoid arthritis. A high coffee intake was linked to both the ability to lower intraocular pressure and a low probability of diagnosing dry eye disease. In infants with respiratory distress syndrome, intake of caffeine (10 and 20 mg/kg daily) was observed to lower nasal continuous positive airway pressure in a randomized controlled trial. Research has shown that caffeine usage may have a minor effect on blood markers linked to cardiovascular diseases.

Also, caffeine is found useful in cosmeceutical preparations (nanoformulations) for treating cellulitis. Additionally, in preclinical studies, caffeine alone or in combination with nanoformulations, such as polymeric nanoparticles, nanospheres, lipid nanoparticles, etc., are effective options for mitigating various ailments, for instance, migraine, liver cancer, breast cancer, cervical cancer, and obesity. Based on meta-analyses and preclinical findings, it can be concluded that moderate consumption of acceptable amounts of caffeinated beverages and caffeine combination products or nano-delivery systems of caffeine may help to improve human life.

## Figures and Tables

**Figure 1 ijms-25-12003-f001:**
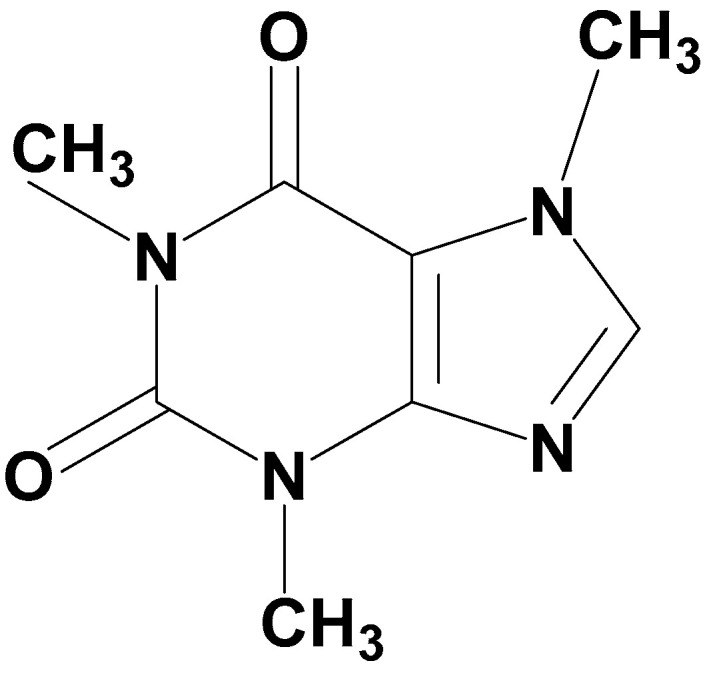
Structure of caffeine.

**Figure 2 ijms-25-12003-f002:**
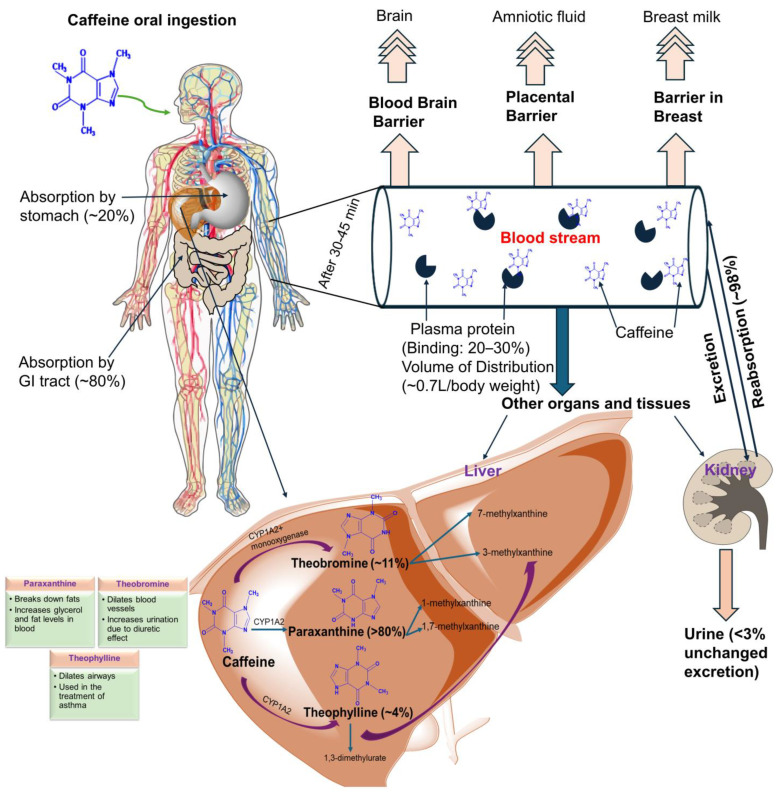
Pharmacokinetics of caffeine and its metabolites.

**Table 1 ijms-25-12003-t001:** Meta-analysis evaluating the relationship between caffeine intake and risk of dementia, AD, and PD.

S. No.	Disease	Country ^†^	n (Sample Size)	Amount of Caffeine Consumption	Conclusions	References
**1**	Dementia and Alzheimer’s disease	Fin	58 subjects of AD, 61 subjects of dementia	Tea and coffee ≤ 5 cups/day	Tea consumption was rather infrequent and had no connection to dementia or AD. Midlife coffee drinking was associated with a lower risk of dementia/AD later in life.	[110]
**2**	Dementia and Alzheimer’s disease	US	118 subjects with AD, 226 subjects with dementia	Caffeine intake: ~100 mg/day–280 mg/day	Midlife consumption of coffee and caffeine was not linked to dementia, cognitive deficits, or certain neuropathologic lesions.	[100]
**3**	Dementia disease	Jp	26 subjects’ dementia	Green tea, coffee, and black tea 1–6 days/week or every day	Black tea or coffee consumption was not linked to dementia incidence, whereas green tea consumption was shown to be strongly linked to lower risk.	[111]
**4**	Dementia disease	Net	814 subjects with dementia (out of 5408 subjects)	Drinking Coffee ~ >3 cups/day	No link found between drinking coffee and dementia.	[112]
**5**	Parkinson’s disease	US	135,916 subjects (47,351 men and 88,565 women)	Caffeine content: 137 mg per cup of coffee, 47 mg per cup of tea, 46 mg per can or bottle of cola beverage, and 7 mg per serving of chocolate candy.	Moderate caffeine use reduces the chance of developing Parkinson’s disease.	[10]
**6**	Parkinson’s disease	US	77,713 women	Lower to high caffeine consumption (1–6 cups of coffee per day)	Caffeine lowers the risk of PD in women who did not take postmenopausal hormones but raises the risk in those who used the hormones (estrogens).	[113]
**7**	Parkinson’s disease	FIN	29,335 subjects (14,293 men and 15,042 women)	The amount of coffee consumed (0, 1–4, and ≥5 cups/day) and drinking ≥ 3 cups of tea/day	Coffee drinking was associated with reducing the risk of PD. More tea drinking was associated with a reduced risk of PD.	[114]
**8**	Parkinson’s disease	US	304,980 subjects	8.4 mg/day to 357 mg/day	In both men and women, caffeine consumption was negatively correlated with the risk of PD. These results imply that there was no difference between the genders in the association between caffeine and PD.	[115]
**9**	Parkinson’s disease	US	184,190 subjects (86,404 men and 97,786 women)	Caffeine content: 137 mg per cup of coffee, 47 mg per cup of tea, 46 mg per can or bottle of cola beverage, and 7 mg per serving of chocolate candy.	Individuals who routinely drink caffeine have a noticeably decreased chance of acquiring PD. Although it was also present in women, this association was notably significant in males.	[116]
**10**	Parkinson’s disease	FIN	6710 subjects (men and women)	1–10 cups of coffee/day	Consuming coffee lowers the risk of Parkinson’s disease; however, the preventive effect of coffee may vary based on other factors.	[117]
**11**	Parkinson’s disease	SG	63,257	Caffeinated beverages (sodas, coffee, black tea, and green tea): 1–3 cups/month and 1–6 cups/day/week	Caffeinated beverages (black tea), demonstrated an inverse association with the risk of Parkinson’s disease.	[118]
**12**	Parkinson’s disease	Tw	Among 13 studies, 9 were grouped into a healthy cohort and the rest into a PD cohort.	Caffeine content: 137 mg per cup of coffee, 47 mg per cup of tea, 46 mg per can or bottle of cola beverage, and 7 mg per serving of chocolate candy.	For both healthy people and people who already had Parkinson’s disease, caffeine altered disease risk and progression.	[119]

^†^ Country: FIN: Finland, US: United States, Jp: Japan, Net: Netherlands, SG: Singapore, Tw: Taiwan.

**Table 2 ijms-25-12003-t002:** Meta-analysis assessing the relationship between caffeine/coffee consumption and risk of different types of cancer.

S. No.	Disease	Country ^†^	n (Sample Size)	Amount of Caffeine Consumption	Conclusions	References
**1**	Breast cancer	Fr	4396 women Subjects	Coffee 112–252 mL/d, Tea 1–349 mL/d, Herbal Tea 1–149 mL/d	Consumption of herbal tea appeared to be associated with a lower risk of breast cancer, although consumption of coffee and tea was not.	[170]
**2**	Breast cancer	Fr	67,703 women	280 mL/d (2.2 cups/d) for coffee, and 214 mL/d (1.7 cups/d) for tea and 164 mg/d Caffeine	There was no association between drinking coffee, tea, or caffeine and the risk of breast cancer in general or by hormone receptor status.	[171]
**3**	Breast cancer	US	85,987 female participants	Coffee, tea, and caffeine, carbonated beverages with caffeine and chocolate 1–4 cup/d	Caffeinated beverages and the risk of postmenopausal breast cancer had a moderate negative association.	[172]
**4**	Breast cancer	EU (Dn, Fr, Gm, Gr, It, Nt, Nw, Sp, Sw, and UK)	335,060 women	Tea and coffee drinkers	Drinking more caffeinated coffee may lower the risk of developing postmenopausal breast cancer. Drinking decaffeinated coffee did not appear to be connected with breast cancer.	[173]
**5**	Breast cancer	Sp	115,802 females (10,812 middle-aged women)	One cup or less of coffee/day	Among postmenopausal women, there was a negative correlation between coffee consumption and the risk of breast cancer.	[174]
**6**	Breast cancer	Cn	37 published articles with 59,018 breast cancer cases	Coffee 2 cups/day, Caffeine 200 mg/day	The risk of breast cancer and coffee did not significantly correlate.	[175]
**7**	Breast cancer	US	57,075 postmenopausal women	Coffee 2 or more cups per day	These findings did not indicate a relationship between coffee intake and postmenopausal women’s risk of developing invasive breast cancer.	[176]
**8**	Hepatocellular cancer (HCC)	Jp	209 incident hepatocellular carcinoma (HCC) cases and 1308 community controls	Coffee 1–2 cups/day or ≥3 cups/day	These results suggested that coffee may prevent the growth of HCC.	[177]
**9**	Hepatocellular cancer (HCC)	HK	234 HBV chronic carriers (109 cases and 125 controls)	Coffee Drinkers ≤ 3 times/week Moderate Coffee Drinkers ≥ 4 times/week	The protective effects of moderate coffee drinking in HBV chronic carriers were supported by the study.	[178]
**10**	Liver cancer	Jp	18,815 subjects	Coffee and green tea 1–3 cups/day or ≥5 ups/day	While there was no relationship between green tea drinking and the risk of liver cancer in all participants, higher coffee consumption was associated with lower risk of liver cancer in all subjects.	[179]
**11**	Hepatocellular cancer (HCC)	Sg	63,257 subjects	Three or more cups of coffee/day	Coffee drinking may lower the risk of HCC in Singapore’s Chinese population.	[180]
**12**	Hepatocellular cancer (HCC)	It	185 cases, 412 controls	Consuming < 14 cups/week of coffee	supports the idea that coffee has a positive effect.	[181]
**13**	Liver cancer	Fn	60,323 subjects	Coffee 1–8 cups/day or ≥cups/day	The risk of liver cancer was inversely and variably correlated with coffee consumption.	[182]
**14**	Liver disease: hepatocellular carcinoma (HCC) and chronic liver disease (CLD)	It	A meta-analysis [12 studies on HCC (3414 cases) and 6 studies on CLD (1463 cases)]	One cup of coffee per day or ≥1 cup/day	There was a negative link between coffee consumption and the risk of HCC, and additional data suggest the presence of a stronger negative association with CLD.	[183]
**15**	Hepatocellular carcinoma (HCC)	UK	88 cases among 471,779 participants	3–4 cups/day or ≥5 cups/day	Coffee consumption showed a negative correlation with HCC.	[184]
**16**	Oral, pharyngeal, and esophageal cancers	JP	47,605 subjects (22,836 men and 24,769 women)	≤4 cups/day or ≥5 cups/day	In the overall Japanese population, coffee intake was associated with a lower incidence of oral, pharyngeal, and esophageal malignancies.	[185]
**17**	Oral or esophageal cancers	Nw	450 squamous oral or esophageal cancers out of 389,624 men and women participants	5–8 cups/day or ≥9 cups/day	There was no evidence to suggest an association between drinking coffee and the risk of developing mouth or esophageal cancer.	[186]
**18**	Esophageal cancer (EC)	EU (Dn, Fr, Gm, Gr, It, Nt, Nw, Sp, Sw, and UK)	Out of 442,143 men and women, 339 developed EC, with 142 developing esophageal adenocarcinoma (EAC) and 174 developing esophageal squamous cell carcinoma (ESC).	Tea 0.1–179 (mL/d) or >179 mL/d, Coffee 150–477 (mL/d) or >477 mL/d	Drinking coffee and tea did not significantly increase the risk of EC or any of its subtypes.	[187]
**19**	Esophageal cancer	Cn	457,010 participants and incident cases of 2628	1–3 cups/day or >7 cups/day	Drinking coffee was shown to protect against esophageal cancer in East Asians but not in Euro-Americans.	[188]
**20**	Gastric cancer	Cn	22 research (9 cohort and 13 case-control studies) included 7631 cases and 1,019,693 controls	Coffee 3–4 cups/day	Studies suggest that there is a link between drinking coffee and a lower incidence of stomach cancer.	[189]
**21**	Gastric cancer	Cn	1,250,825 participants and 3027 cases of gastric cancer	Coffee drinkers never, 1 cup, once, or 131 mL per day	Coffee drinking was not associated with an increased risk of gastric cancer in general; however, it may be associated with a higher risk of gastric cardia cancer.	[190]
**22**	Gastric cancer	Cn	312,993 volunteers (among them 1429 were diagnosed with gastric cancer)	Not available	Gastric cancer incidence was correlated with coffee drinking. The risk of stomach cancer may rise with increased coffee consumption.	[191]
**23**	Gastric cancer	Cn	3484 gastric cancer patients out of 1,324,559 participants	Coffee drinker 1–7 cups/day or ≥10 cups/day	A high coffee consumption increases the chance of developing stomach cancer.	[192]
**24**	Pancreatic cancer	US	1541 new pancreatic cancers among 457,366	Coffee (caffeinated, decaffeinated)	There was no link between drinking coffee overall, whether caffeinated, or decaffeinated, and developing pancreatic cancer.	[193]
**25**	Pancreatic cancer	Fn	235 cases of pancreatic cancer out of 60,041 subjects	Coffee drinkers 1–8 cups/day or ≥10 cups/day	No significant link between drinking coffee and the risk of pancreatic cancer.	[194]
**26**	Cancers (oral, pharynx, liver, colon, prostate, endometrial cancer and melanoma and lung cancer)	Cn	1,395,309 samples	Coffee drinker 2 cups/day	Coffee consumption was linked to an increased risk of lung cancer and a lower risk of cancers of the mouth, throat, liver, colon, prostate, endometrium, and melanoma.	[195]
**27**	Colorectal cancer	US	681 colorectal cancer cases out of 57,398 men and women participants	Coffee and tea drinker	This study did not support any claims that consuming coffee or tea can help prevent colorectal cancer.	[196]
**28**	Colorectal cancer	US	83,778 women participants	Coffee drinkers None, 1–4 cups/day or ≥4 cups/day	Increased coffee intake was linked to an increase in colorectal cancer incidence.	[197]
**29**	Colorectal cancer	Pl	3,402,167 participants	Coffee drinkers 1 cup/day or ≥2 cups/day	No evidence that consuming coffee increases the risk of developing CRC.	[198]
**30**	Colorectal cancer	US	1829 cases out of 47,010 men and 60,051 women	≥2 cups/day of caffeinated coffee or ≥6 cups/day	Higher intake of decaffeinated coffee was linked to a decreased risk of colorectal cancer, but higher consumption of caffeinated coffee was linked to a higher risk.	[199]
**31**	Kidney cancer	US	318 kidney cancers out of 96,024 individuals	Coffee and tea or caffeine drinkers ≥2–4 cups/day	One cup per day consumption was linked to a lower incidence of kidney cancer.	[200]
**32**	Urologic cancer [prostate cancer (PCa), bladder and kidney cancer]	Cn	969 bladder cancer cases out of 201,272 participants, 32,735 PCa cases out of 447,458 participants, 366 kidney cancer out of 310,625 participants	Coffee drinkers 2–4 cups/day Or ≥4	Drinking coffee may reduce the risk of PCa. Both bladder and kidney cancer did not show any associations.	[201]
**33**	Bladder cancer [urothelial cell carcinomas (UCC)]	EU	513 UCC cases among 233,236 subjects	Beverages included coffee (250–500 mL/day or ≥500 mL/day), tea and herbal tea (200 mL/day or ≥250 mL/day and others.	There were no links found between drinking coffee, tea, or herbal tea with the risk of UCC.	[202]
**34**	Cancers and subgroups of cancers (bladder, breast, buccal and pharyngeal, colorectal, endometrial, esophageal, hepatocellular, leukemic, pancreatic, and prostate cancers)	Cn	34,177 incident cases of cancer out of 2,179,126 participants	Coffee drinkers 1–6 cups/day	Coffee drinking may reduce overall cancer incidence and has a negative association with certain cancer types (bladder, breast, buccal, pharyngeal, colorectal, endometrial, esophageal, hepatocellular, leukemic, pancreatic, and prostate cancers).	[203]
**35**	Bladder cancer	Cn	700 disease cases out of 229,099 participants	Coffee drinkers 1–4 cups/day	No evidence that drinking coffee increases the risk of bladder cancer.	[204]
**36**	Cancers (head and neck, esophageal, stomach, lung, breast, endometrial, ovarian, prostate, kidney, bladder, gliomas, and thyroid)	US	Certain cancer incidence of 145 head and neck, 99 esophageal, 136 stomach, 1137 lung, 1703 breast, 257 endometrial, 162 ovarian, 3037 prostate, 318 kidney, 398 bladder, 103 gliomas, and 106 thyroids	Consumption of coffee, tea, and caffeine	While tea drinking was associated with a decreased risk of cancer overall, coffee drinking was not associated with a lower risk of endometrial cancer and an increased risk of other cancers. A dose–response association between caffeine use and cancer risk was not found.	[200]
**37**	Prostate cancer	Sw	Out of 406,718 subjects, 5733 cases of low-grade PCa, 25,188 cases of localized PCa, 1965 cases of high-grade PCa, and 5724 cases of advanced PCa	Coffee drinkers ≥3 cups/day	The risk of fatal PCa may be negatively correlated with coffee intake. There was no strong evidence of a relationship with PCa incidence.	[205]
**38**	Prostate cancer	Cn	34,105 cases of prostate cancer out of 539,577 participants	Coffee drinkers ≥ 2 cups/day	Drinking coffee may be associated with a decreased risk of prostate cancer.	[206]
**39**	Prostate cancer	UK	46,687 men	Coffee/tea drinkers ≥4 cups/day	Coffee drinking was related to a reduced risk for prostate cancer, which could be due other to confounding lifestyle factors. According to findings, coffee drinking does not directly influence prostate cancer incidence.	[207]
**40**	Ovarian cancer	UK, Dn, Sw, Nt, Gm, Fr, Sp, Nw, Gr, Au	Among 330,849 women, 1216 had ovarian cancer, 43 with peritoneal tumors, and 49 with fallopian tube tumors.	Coffee and tea consumption ≥1 cup/day	No evidence that drinking coffee or tea increases the chance of developing ovarian cancer.	[208]
**41**	Ovarian cancer	Irn	4434 ovarian cancer cases out of 940,359 participants	Coffee drinkers 1 cup/day or ≥1 cup/day	No link between caffeine intake or the drinking of caffeinated or decaffeinated coffee with the incidence of ovarian cancer.	[209]

**^†^ Country:** Fr: France, Sp: Spain, Cn: China, US: United States, Jp: Japan, Dn: Denmark, UK: United Kingdom, Sw: Sweden, Nw: Norway, HK: Hong Kong, Nt: Netherlands, Gm: Germany, Gr: Greece, Pl: Poland, Au: Austria, It: Italy, Irn: Ireland, EU: European Union.

**Table 3 ijms-25-12003-t003:** Meta-analysis assessing the relationship between caffeine/coffee consumption and diabetes risk.

S. No.	Disease	Country ^†^	n (Sample Size)	Amount of Caffeine Consumption	Conclusions	References
**1**	Diabetes: T1DM	UK	34 patients (22 males, 12 females)	200 mg caffeine capsules twice a day	Men were more susceptible to the effects of caffeine than women. Caffeine increased the severity of hypoglycemia warning signals in type 1 diabetic patients.	[215]
**2**	Diabetes: T1DM	UK	15 subjects 5 males, 10 females	Energy drinks 240 mg	Blood glucose levels were not raised by caffeine alone.	[216]
**3**	Diabetes: T1DM	Ca	13 subjects (5 males, 8 females)	Capsules 6 mg/kg body weight	Caffeine consumption increased blood glucose levels during exercise, while the levels dramatically decreased after exercise.	[217]
**4**	Diabetes: T1DM	UK	19 subjects (9 males, 10 females)	Capsules 250 mg/day twice	Coffee consumption lowered overnight hypoglycemia, but there was no difference in average glycemic control.	[218]
**4**	Diabetes: T2DM	NZ	18 subjects (9 males, 9 females)	Espresso coffee beverage ~180 mg	OGTT (oral glucose tolerance test) results showed a little rise in blood sugar following coffee consumption, but no change in insulin sensitivity.	[219]
**5**	Diabetes: T2DM	Ca	12 male subjects	Capsules 5 mg/kg body weight	Caffeine consumption increased BG levels after the OGTT and reduced episodes of hypoglycemia.	[220]
**6**	Diabetes: T2DM	Br	8 subjects	Caffeine beverage 1.5 mg/kg body weight	No difference in BG during exercise was seen with caffeine consumption.	[221]
**7**	Pre-diabetes and T2D	Iran	1878 adults (844 men, 1034 women) and 2139 adults (971 men, 1168 women) with an age of 20–70	Caffeine (≥152 vs. <65 mg/d)	Caffeine from a normal diet may have a positive effect on prevention of pre-diabetic by modifying insulin and insulin resistance	[212]
**8**	Type 2 diabetes	USA	45 health cohort studies	5 cups (177 mL each) of instant caffeinated coffee, decaffeinated coffee/day for 8 weeks	Negatively associated with the risk of type 2 diabetes	[213]

**^†^ Country:** UK: United Kingdom, Ca: Canada, Br: Brazil, NZ: New Zealand.

**Table 4 ijms-25-12003-t004:** Examples of various nanoformulations developed using caffeine alone or in combination.

Formulations	Active(s)	Parameters *	Methods	Disease/Target	Outcomes	References
**Nano-Chitosome**	Caffeine	PS: 135.5–533.5 nm, PSD: 0.31–0.41, ZP: +40.96 to +50.5	Thin-film hydration	Taste improvement	Improved taste by masking bitterness	[260]
**Chitosan nanoparticles**	Alpha-lipoic acid (α-LA) and caffeine	--	---	Cardiovascular	Reduce the negative consequences of obesity on the heart; AchE, MAO, Na+/K+-ATPase, and monoamines were restored.	[261]
**Chitosan nanoparticles**	Alpha-lipoic acid (α-LA) and caffeine	Average PS: 4–12 nm	Gelation followed by cross-linking	Obesity treatment, effect on liver and kidney	Significant reduction in renal Na+, K+-ATPase activity and hepatic and renal GSH	[262]
**Solid lipid nanoparticles (SLN)**	Chlorogenic acid and caffeine	PS: 110.2 ± 0.1 nm	HPH	Anti-obesity	SLN showed a better effect (*p* < 0.05) on the adipocyte cells	[263]
**Silver nanoparticle (AgNPs)**	Caffeine	PS: 16–24 nm	Green synthesis	Antimicrobial	Successfully prevented *E. coli* and *S. aureus* growth	[264]
**Vesosomes**	Ergotamine and caffeine	PS: 315.48 ± 14.27 nm ZP: −21.78 ± 4.72	Ethanol-induced interdigitation of lipids	Antimigraine	Better absorption and bioavailability	[265]
**Hybrid nanoparticles**	Ergotamine and caffeine	PS: 239.46 ± 2.31 nm ZP: −18.36 ± 6.59 mV	Nanoprecipitation method	Antimigraine	Showed synergistic activity with good anti-migraine potential	[266]
**Chitosan nanoparticles**	Caffeine and folic acid	PS: 140 ± 95.65 nm, ZP: 16.6 ± 3.46 mV	Ionic gelation method	Liver cancer	Showed antioxidant and cytotoxicity activity in different cancer cells	[267]
**Nano-immunomodulator**	R837 and caffeine	R-nIMs PS: 150 ± 6.5, ZP: −19.5 ± 0.6 RC-nIMs PS: 154 ± 5.8, ZP: −23.4 ± 1.1	HPH	Orthotopic breast cancer	Exhibited good antitumor effects	[149]
**MgONPs coated w/o Pickering emulsion**	Caffeine	PS: 94.6 ± 19	Sol–gel and Homogenization	Hepatocellular carcinoma	Significant hepatoprotection and maintained the effective sustained level of caffeine	[268]
**Gelatin nanoparticles**	Caffeine	PS: 480 and 870 nm	Two-step desolvation	Melanoma	Inhibited cell viability and migration ability of melanoma cells	[269]
**Mesoporous silica nanoparticles**	Caffeine	--	--	Anti-leukemic potential	Excellent therapeutic agent against AML	[270]
**Microemulsified system**	Caffeine	--	--	Breast cancer	Demonstrating the antitumor activity via oxidative stress in MCF7 cells	[271]
**Self-assembled nanospheres**	Caffeine and doxorubicin	PS: 155 nm	--	Cervical cancer	Safe and efficient intracellular reduction stimulus-responsive drug-delivery vehicles against HeLa cells	[147]

* PS: particle size, ZP: zeta potential, MAO: monoamine oxidase, AchE: acetylcholinesterase, HPH: High-pressure homogenizer, R-nIMs: R837 loaded nano-immunomodulators, RC-nIMs: R837 and caffeine loaded nano-immunomodulators, AML: acute myeloid leukemia.

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
