# Peer review of "Caffeine: A Multifunctional Efficacious Molecule with Diverse Health Implications and Emerging Delivery Systems"

_ijms, 2024, doi:10.3390/ijms252212003_

Round 1
Reviewer 1 Report
Comments and Suggestions for Authors
This comprehensive review thoroughly examines caffeine's multifaceted effects on human health, focusing on its neuroprotective, anticancer, and metabolic properties. Additionally, it highlights the advancements in caffeine delivery systems, particularly nanoformulations, to improve bioavailability and therapeutic outcomes.
Minor:
1. Change the style of citation
2. Authors should add caffeine structure in the introduction section after the chemical formula
3. Line 631,634 – missing space
4. Table 3 – The notation of countries should be unified, either all written with their full name or abbreviation
Major:
1. Figure 1 – some text in the figure is unreadable; its quality should be improved
2. Due to the length of the review itself and its complexity, the conclusions section should be more extensive to summarize all the topics covered fully
Incorporating the changes requested will allow this review to be ready for acceptance.
Comments on the Quality of English Language
Minor stylistic and linguistic corrections are required
Author Response
Reviewer 1
This comprehensive review thoroughly examines caffeine's multifaceted effects on human health, focusing on its neuroprotective, anticancer, and metabolic properties. Additionally, it highlights the advancements in caffeine delivery systems, particularly nanoformulations, to improve bioavailability and therapeutic outcomes.
Minor:
- Change the style of citation
Response : Thank you for highlighting the error. The authors have made the corrections in the revised manuscript.
- Authors should add caffeine structure in the introduction section after the chemical formula
Response : Thank you for your useful recommendation. The authors have made the corrections in the revised manuscript.
- Line 631,634 – missing space
Response : Thank you for your suggestion. The authors have done the corrections in the revised manuscript.
- Table 3 – The notation of countries should be unified, either all written with their full name or abbreviation
Response : Thank you for your useful suggestion. The authors have corrected the revised manuscript.
Major:
- Figure 1 – some text in the figure is unreadable; its quality should be improved
Response : Thank you for your helpful suggestion. The authors have made corrections in the revised manuscript.
- Due to the length of the review itself and its complexity, the conclusions section should be more extensive to summarize all the topics covered fully
Response : Thank you for your helpful suggestion. The authors have made corrections in the revised manuscript.
Incorporating the changes requested will allow this review to be ready for acceptance.
Response : Thank you for your recommendation for accepting the manuscript after revision. The authors have made corrections carefully in the revised manuscript, as per comments and suggestions.
Comments on the Quality of English Language: Minor stylistic and linguistic corrections are required
Response : Thank you for your useful suggestion. The authors have corrected the revised manuscript.

Reviewer 2 Report
Comments and Suggestions for Authors
- The introduction covers several important points about caffeine’s properties, health benefits, and technological innovations, but the transitions between ideas could be smoother. For example, the shift from caffeine’s health benefits to the discussion of nano/micro platforms feels abrupt. Consider adding linking sentences to improve the flow between these sections.
- Additionally, the sentence beginning with "Coffee is probably the most popular beverage..." (lines 40-42) could benefit from rephrasing for clarity. While the meaning is clear, the structure could be simplified for better readability.
- Line 43: Instead of "less toxic," a more specific phrase like "relatively low toxicity" would be more accurate and scientific.
- When discussing the health impacts of caffeine, phrases like "modulation of reactive oxygen species production" (line 55) could benefit from brief explanations or citations to clarify the mechanism for readers who may not be familiar with this concept.
- The use of citations is generally appropriate, but ensure that each claim, especially those about health effects and technological applications (lines 54-60), is backed by relevant references. The reader may want to see more direct references, particularly for statements about the negative effects of caffeine on athletes, pregnant women, and its impact on body temperature (line 58).
- Additionally, the introduction touches on many points briefly (caffeine’s benefits, consumption patterns, technological solutions), but could delve deeper into each before moving on. A more detailed exploration of caffeine’s adverse effects and why novel delivery methods are necessary would strengthen the argument for the rest of the review.
- The explanation of the "first-pass effect" (lines 71-73) is a bit confusing. While it is correct that caffeine has a minimal hepatic first-pass effect, this could be rephrased for clarity, especially for readers unfamiliar with pharmacokinetics.
- In lines 85-87, you mention the plasma concentration levels following different doses of caffeine. It would be helpful to briefly mention the clinical relevance of these concentrations, especially when discussing typical doses from coffee (0.4–2.5 mg/kg) and peak plasma concentrations.
- The description of caffeine metabolism by cytochrome P450 enzymes is well done, but Figure 1 could be referenced more clearly when discussing the breakdown into paraxanthine, theophylline, and theobromine (lines 105-109). This would help readers follow along with the metabolic pathways being described.
-
Caffeine and Migraine:
- This section starts with a good explanation of caffeine’s use in migraine treatment and the challenges of oral caffeine delivery. However, you could improve the discussion by explaining the physiological reasons behind these challenges, such as the pharmacokinetic limitations (e.g., short half-life and gastric emptying effects).
- Additionally, the introduction of the transdermal patch (PNS-4) study is interesting but feels abrupt. You could introduce the rationale for this delivery method earlier, perhaps before discussing the clinical trial, and give more detail on the patch technology itself. Why does transdermal administration result in better brain targeting compared to oral caffeine delivery?
- The discussion of ALS could be more focused on the mechanisms behind caffeine’s neuroprotective effects, particularly how A2A adenosine receptor (A2AAR) antagonism plays a role. You mention this, but it would help to give a bit more context about how the A2AAR receptor is involved in neurodegeneration.
- Similarly, the section on MS highlights relevant studies but could use more explanation of the biochemical pathways involved, such as how the upregulation of A1AR by caffeine reduces MS symptoms. A clearer explanation of these receptor functions would make the pharmacological mechanisms more accessible to readers.
- The section on AD provides valuable information on how caffeine interferes with amyloid beta (Aβ) aggregation, an important aspect of AD pathology. However, the explanation of how caffeine inhibits β- and γ-secretases (lines 170-172) could be expanded with a brief explanation of these enzymes’ roles in Aβ production and accumulation.
- The introduction of caffeine’s effect on A2AAR and A3AR receptors is well done but might benefit from a sentence or two summarizing why these receptors are critical in AD. The transition between discussing Aβ and A2AAR could also be smoother.
- Additionally, it would be useful to explain the significance of the results of caffeine’s protective effect in AD models and how these preclinical findings relate to potential clinical applications.
-
- The PD section is strong, providing a good overview of caffeine’s neuroprotective role, particularly through its effect on the dopaminergic system and the blocking of A2AAR receptors. The section flows well, but the relationship between GRIN2A genotype and PD progression (lines 199-202) could use more explanation. Why is the GRIN2A T allele relevant, and how does it affect caffeine’s impact on PD?
- The discussion of caffeine’s antioxidant effects is very informative. However, I would recommend briefly expanding on the specific antioxidant mechanisms (e.g., glutathione reductase and superoxide dismutase activity) to give the reader a clearer understanding of how caffeine might reduce oxidative stress in PD.
- Effect on Huntington's Disease: There's a mention of contradictory findings between epidemiological studies and animal models, but this point is not fully explored. It would be beneficial to delve deeper into why these contradictions exist or suggest directions for future research.
Author Response
Reviewer 2
Comments and Suggestions for Authors
- The introduction covers several important points about caffeine’s properties, health benefits, and technological innovations, but the transitions between ideas could be smoother. For example, the shift from caffeine’s health benefits to the discussion of nano/micro platforms feels abrupt. Consider adding linking sentences to improve the flow between these sections.
Response : Thank you for your positive comments and suggestions to improve the manuscript. The authors have incorporated the suggestions to the revised manuscript.
- Additionally, the sentence beginning with "Coffee is probably the most popular beverage..." (lines 40-42) could benefit from rephrasing for clarity. While the meaning is clear, the structure could be simplified for better readability.
Response : Thank you for your valuable suggestion. The authors have been made the correction in the revised manuscript.
- Line 43: Instead of "less toxic," a more specific phrase like "relatively low toxicity" would be more accurate and scientific.
Response : Thank you for your efficient suggestion. The authors have been made the correction in the revised manuscript.
- When discussing the health impacts of caffeine, phrases like "modulation of reactive oxygen species production" (line 55) could benefit from brief explanations or citations to clarify the mechanism for readers who may not be familiar with this concept.
Response : Thank you for your useful recommendation. The authors have been made the correction in the revised manuscript.
- The use of citations is generally appropriate, but ensure that each claim, especially those about health effects and technological applications (lines 54-60), is backed by relevant references. The reader may want to see more direct references, particularly for statements about the negative effects of caffeine on athletes, pregnant women, and its impact on body temperature (line 58).
Response : Thank you for your useful suggestion. The authors have made the corrections in the revised manuscript.
- Additionally, the introduction touches on many points briefly (caffeine’s benefits, consumption patterns, technological solutions), but could delve deeper into each before moving on. A more detailed exploration of caffeine’s adverse effects and why novel delivery methods are necessary would strengthen the argument for the rest of the review.
Response : Thank you for your useful suggestion. The required information has been added to the revised manuscript.
- The explanation of the "first-pass effect" (lines 71-73) is a bit confusing. While it is correct that caffeine has a minimal hepatic first-pass effect, this could be rephrased for clarity, especially for readers unfamiliar with pharmacokinetics.
Response : Thank you for your useful suggestion. The authors have been made the correction in the revised manuscript.
- In lines 85-87, you mention the plasma concentration levels following different doses of caffeine. It would be helpful to briefly mention the clinical relevance of these concentrations, especially when discussing typical doses from coffee (0.4–2.5 mg/kg) and peak plasma concentrations.
Response : Thank you for your useful suggestion. The authors have been made the correction in the revised manuscript.
- The description of caffeine metabolism by cytochrome P450 enzymes is well done, but Figure 1 could be referenced more clearly when discussing the breakdown into paraxanthine, theophylline, and theobromine (lines 105-109). This would help readers follow along with the metabolic pathways being described.
Response : Thank you for your useful suggestion. The authors have made the corrections in the revised manuscript.
- Caffeine and Migraine:
- This section starts with a good explanation of caffeine’s use in migraine treatment and the challenges of oral caffeine delivery. However, you could improve the discussion by explaining the physiological reasons behind these challenges, such as the pharmacokinetic limitations (e.g., short half-life and gastric emptying effects).
Response : Thank you for your useful suggestion. The required information has been added to the revised manuscript.
- Additionally, the introduction of the transdermal patch (PNS-4) study is interesting but feels abrupt. You could introduce the rationale for this delivery method earlier, perhaps before discussing the clinical trial, and give more detail on the patch technology itself. Why does transdermal administration result in better brain targeting compared to oral caffeine delivery?
Response : Thank you for your helpful suggestion to improve the manuscript. The authors have corrected the revised manuscript.
- The discussion of ALS could be more focused on the mechanisms behind caffeine’s neuroprotective effects, particularly how A2A adenosine receptor (A2AAR) antagonism plays a role. You mention this, but it would help to give a bit more context about how the A2AAR receptor is involved in neurodegeneration.
Response : Thank you for your useful suggestion. The required information has been added in the revised manuscript.
- Similarly, the section on MS highlights relevant studies but could use more explanation of the biochemical pathways involved, such as how the upregulation of A1AR by caffeine reduces MS symptoms. A clearer explanation of these receptor functions would make the pharmacological mechanisms more accessible to readers.
Response : Thank you for your useful suggestion. The required information has been added to the revised manuscript.
- The section on AD provides valuable information on how caffeine interferes with amyloid beta (Aβ) aggregation, an important aspect of AD pathology. However, the explanation of how caffeine inhibits β- and γ-secretases (lines 170-172) could be expanded with a brief explanation of these enzymes’ roles in Aβ production and accumulation.
Response : Thank you for your valuable recommendation. The required information has been added to the revised manuscript.
- The introduction of caffeine’s effect on A2AAR and A3AR receptors is well done but might benefit from a sentence or two summarizing why these receptors are critical in AD. The transition between discussing Aβ and A2AAR could also be smoother.
Response : Thank you for your useful suggestion. The required information has been added to the revised manuscript.
- Additionally, it would be useful to explain the significance of the results of caffeine’s protective effect in AD models and how these preclinical findings relate to potential clinical applications.
Response : Thank you for your useful suggestion. The authors have been made the correction in the revised manuscript.
- The PD section is strong, providing a good overview of caffeine’s neuroprotective role, particularly through its effect on the dopaminergic system and the blocking of A2AAR receptors. The section flows well, but the relationship between GRIN2A genotype and PD progression (lines 199-202) could use more explanation. Why is the GRIN2A T allele relevant, and how does it affect caffeine’s impact on PD?
Response : Thank you for your useful suggestion. The required information has been added to the revised manuscript.
- The discussion of caffeine’s antioxidant effects is very informative. However, I would recommend briefly expanding on the specific antioxidant mechanisms (e.g., glutathione reductase and superoxide dismutase activity) to give the reader a clearer understanding of how caffeine might reduce oxidative stress in PD.
Response : Thank you for your valuable suggestion. The required information has been added to the revised manuscript.
- Effect on Huntington's Disease:There's a mention of contradictory findings between epidemiological studies and animal models, but this point is not fully explored. It would be beneficial to delve deeper into why these contradictions exist or suggest directions for future research.
Response : Thank you for your helpful suggestion. The required information has been added to the revised manuscript.
Caffeine intake of up to 400 mg per day in healthy people is largely supported by the data to be free of conspicuously harmful cardiovascular effects, behavioral effects, reproductive and developmental impacts, acute effects, and bone state [1]. Further, there is evidence that pregnant women in good health can consume up to 300 mg of caffeine per day, an amount that is typically not linked to negative effects on reproduction and development. For children and adolescents, there was little data found; nonetheless, the information that is now available indicates that 2.5 mg of caffeine per kilogram of body weight per day is still a suitable prescription.
- Wikoff, D.; Welsh, B. T.; Henderson, R.; Brorby, G. P.; Britt, J.; Myers, E.; Goldberger, J.; Lieberman, H. R.; O'Brien, C.; Peck, J., Systematic review of the potential adverse effects of caffeine consumption in healthy adults, pregnant women, adolescents, and children. Food and chemical toxicology 2017, 109, 585-648.

Round 2
Reviewer 2 Report
Comments and Suggestions for Authors.